# Development and External Validation of a Radiomics Model Derived from Preoperative Gadoxetic Acid-Enhanced MRI for Predicting Histopathologic Grade of Hepatocellular Carcinoma

**DOI:** 10.3390/diagnostics13030413

**Published:** 2023-01-23

**Authors:** Xiaojun Hu, Changfeng Li, Qiang Wang, Xueyun Wu, Zhiyu Chen, Feng Xia, Ping Cai, Leida Zhang, Yingfang Fan, Kuansheng Ma

**Affiliations:** 1The Department of General Surgery & Hepatobiliary Surgery, Zhujiang Hospital, Southern Medical University, Guangzhou 510280, China; 2Department of Hepatobiliary Surgery, The Fifth Affiliated Hospital of Southern Medical University, Guangzhou 510920, China; 3Institution of Hepatobiliary Surgery, Southwest Hospital, Army Medical University, Chongqing 400038, China; 4Division of Medical Imaging and Technology, Department of Clinical Science, Intervention and Technology (CLINTEC), Karolinska Institutet, 14152 Stockholm, Sweden; 5Department of Radiology, Karolinska University Hospital Huddinge, 14186 Stockholm, Sweden; 6Department of Radiology, Southwest Hospital, Army Medical University, Chongqing 400038, China

**Keywords:** radiomics, magnetic resonance imaging, pathologic grade, gadoxetic acid, hepatocellular carcinoma, machine learning

## Abstract

Histopathologic grade of hepatocellular carcinoma (HCC) is an important predictor of early recurrence and poor prognosis after curative treatments. This study aims to develop a radiomics model based on preoperative gadoxetic acid-enhanced MRI for predicting HCC histopathologic grade and to validate its predictive performance in an independent external cohort. Clinical and imaging data of 403 consecutive HCC patients were retrospectively collected from two hospitals (265 and 138, respectively). Patients were categorized into poorly differentiated HCC and non-poorly differentiated HCC groups. A total of 851 radiomics features were extracted from the segmented tumor at the hepatobiliary phase images. Three classifiers, logistic regression (LR), support vector machine, and Adaboost were adopted for modeling. The areas under the curve of the three models were 0.70, 0.67, and 0.61, respectively, in the external test cohort. Alpha-fetoprotein (AFP) was the only significant clinicopathological variable associated with HCC grading (odds ratio: 2.75). When combining AFP, the LR+AFP model showed the best performance, with an AUC of 0.71 (95%CI: 0.59–0.82) in the external test cohort. A radiomics model based on gadoxetic acid-enhanced MRI was constructed in this study to discriminate HCC with different histopathologic grades. Its good performance indicates a promise in the preoperative prediction of HCC differentiation levels.

## 1. Introduction

Hepatocellular carcinoma (HCC) ranks the sixth most common cancer and the third leading cause of cancer-related mortality worldwide [1]. Although significant advances have been achieved in surgical resection, radiofrequency ablation, and liver transplantation in recent years, high recurrence after these curative treatments remains a big challenge for the long-term survival of HCC patients [2]. The reported recurrence rate reaches as high as 70% at 5 years [2]. Histopathologic grade of HCC, which reflects the tumor’s biological behavior, is one of the key predictors of the prognosis [3]. Compared with moderately or well-differentiated HCC, poorly differentiated HCC is indicative of an early recurrence and lower overall survival [3]. Therefore, for HCC patients with poor differentiation, an expanded resection margin of the tumor during the operation or adjuvant treatments (such as interventional therapy) may be needed to reduce the risk of early recurrence and to improve the outcomes [4,5,6]. In addition, frequent surveillance after the treatments for patients with poorly differentiated HCC is also recommended [6]. Previous research did not recommend liver transplantation in patients with poorly differentiated HCC > 3 cm [4]. 

To date, the only approach to obtain the tumor grading information preoperatively is via biopsy. However, this approach is invasive, which leaves the patients at risk of post-procedural complications, such as bleeding, tumor seeding, and pain, with a rate of around 6% [7]. In addition, false negative results may occur due to sampling errors in the biopsy [8]. Moreover, the limited sample obtained by biopsy may not thoroughly reflect the intra-tumoral heterogeneity. Preoperative HCC differentiation information is an unmet clinical requirement for clinicians. 

Gadoxetic acid-enhanced magnetic resonance imaging (MRI) has been increasingly used in clinical practice for the detection, diagnosis, and characterization of focal and diffuse hepatic disease [9]. Gadoxetic acid is a liver-specific contrast media, indicating that it not only distributes into the extracellular space after intravenous injection but also can be actively taken up by the hepatocytes via the organic anion transporting polypeptides (OATPs) [10]. Up to 50% of administrated gadoxetic acid will be excreted into the bile duct via multidrug resistance protein 2 [10]. The uptake of gadoxetic acid reaches a peak at 10–40 min after injection, forming a specific hepatobiliary phase [11]. Normal liver parenchyma usually enhances over time due to hepatocyte uptake of gadoxetic acid, but HCC typically appears as hypointense nodules [9]. Due to the better tissue contrast provided by the hepatobiliary phase, gadoxetic acid-enhanced MRI improves the detection rate and diagnostic accuracy of HCC [10]. In recent years, studies have shown that radiological features at gadoxetic acid-enhanced MRI, such as intratumoral vessels, and peritumoral hypointensity, are associated with HCC histopathologic grading results [12,13]. However, these semantic features are limited by subjectivity and low accuracy. 

Radiomics is a novel technique that enables quantitative analysis of medical images beyond visual inspection by extracting high-throughput textual features and converting them into mineable data [14,15]. Taking advantage of machine learning approaches, the radiomics model can be developed to improve the accuracy of diagnosis and prediction of treatment response and prognosis, carrying the potential to assist clinical decision-making [16]. In the field of liver oncology, several radiomics models have been constructed using gadoxetic acid-enhanced MRI for HCC histopathologic grading prediction, and they showed excellent discriminative ability [17,18,19]. However, the generalizability of the developed model remains unclear as these models were based on limited sample size and lacked external validation. In this study, we aimed to develop a radiomics model for preoperatively predicting the HCC histopathologic grade using gadoxetic acid-enhanced MRI and to validate it in an independent external cohort. 

## 2. Patients and Methods 

The ethical review boards of Southwest Hospital, Army Medical University, Chongqing (Center 1), and Zhujiang Hospital, South Medical University, Guangzhou (Center 2) approved the protocol of this bi-institutional research [No. (B)KY2022183 and 2022-KY-200-01, respectively]. The written informed consent was waived from the patients due to the retrospective nature of this study. The patient and imaging data were analyzed anonymously. 

Consecutive patients who underwent liver resection between January 2017 and March 2019 at Center 1, and between February 2015 and December 2020 at Center 2 were included in this study if they satisfied the following inclusion criteria: 1. Solitary HCC was confirmed by postoperative pathology exam; 2. Gadoxetic acid-enhanced MRI exam was performed within 2 weeks before surgery; 3. HCC histopathologic grade information was available. Exclusion criteria: 1. Other anti-cancer treatments before surgery, such as radiofrequency ablation, transarterial chemoembolization, and hepatectomy; 2. Incomplete information on clinicopathological variables and pathology report; 3. Other concurrent cancer; 4. Imaging quality insufficient for analysis, such as motion artifact. The process of patient selection is described in Figure 1.

Center 1 was used for model construction (i.e “model development cohort”) and Center 2 served as an external cohort (i.e., “test cohort”) for validating the predictive performance of the developed radiomics model. The test cohort was independent of the model development process. Center 1 or the development cohort, and Center 2 or the test cohort will be interchangeably used in this paper. Figure 2 illustrates the study pipeline of radiomics model development. 

### 2.1. Clinicopathological Variables 

The following preoperative clinicopathological variables were collected from the hospital electronic health records: age (≤55 or >55 years), gender (male/female), hepatitis B virus infection (yes/no), cirrhosis (yes/no), alanine aminotransferase (ALT) (≤42 or >42 IU/L), aspartate transaminase (AST) (≤42 or >42 IU/L), platelet count (≤125 or >125 × 10^9^/L), Albumin-Bilirubin (ALBI) grade (a compound parameter derived from serum albumin and bilirubin levels) (Grade A or B/C) [20], the model for end-stage liver disease (MELD) score (≤9 or >9), tumor size (defined as the maximal diameter at the gadoxetic acid-enhanced MRI) (≤5 or >5 cm) and the alpha-fetoprotein (AFP) level (<400 or ≥400 ng/mL). 

HCC histopathologic grade was defined at the resected specimens by using the World Health Organization criteria, which categorize HCC into three grades: poor, moderate, and well differentiation [21]. In this study, we merged the latter two and grouped patients into two groups: the poorly differentiated HCC group and the non-poorly differentiated HCC group. 

### 2.2. Gadoxetic Acid-Enhanced MRI Exam

All patients underwent the gadoxetic acid-enhanced MRI exam at a 3.0 T scanner (Magnetom Trio, Siemens Healthcare, Erlangen, Germany at Center 1, and Ingenia, Philips Healthcare, Best, The Netherlands at Center 2). Dynamic contrast-enhanced images were acquired before and at the time of aorta enhancement (18–20 s), 45–60 s, 180 s, and 15 min (20 min at Center 2) after administration of the contrast media, which were corresponding to arterial phase, portal venous phase, delayed phase, and hepatobiliary phases. Gadoxetic acid (Primovist ^®^, Bayer Pharma, Berlin, Germany) at a concentration of 0.25 mmol/mL was injected by 0.1 mL/kg body weight through an antecubital vein followed by a flush of saline. Detailed scanning parameters at the two participant centers are provided in Appendix A. Due to its superb tissue contrast, hepatobiliary phase images were selected to develop the radiomics model in this study. 

### 2.3. Tumor Segmentation and Inter-Observer Agreement Assessment

Tumor segmentation was performed manually by two researchers (C.L, 3 years of abdominal imaging research experience, confirmed by P.C, a senior radiologist with 20 years of experience) at the hepatobiliary phase images using the software ITK-SNAP (version 3.8.0, http://www.itksnap.org/, access on 8 August 2022). The delineated volume of interest (VOI) was used to extract the radiomics features. To evaluate the reproducibility of the extracted radiomics features, the interclass coefficient (ICC) was calculated using 30 randomly selected sets of images at Center 1 which were delineated independently by two researchers (C.L and P.C) without the knowledge of the histopathologic results. Radiomics features with ICC ≥ 0.75 were regarded as reproducible and selected for next-step analysis. 

### 2.4. Feature Extraction

To combat the “center effect” of the radiomics features between the two participating hospitals which was potentially derived from the different vendors, scanning parameters, and phase timing used, three strategies were adopted: first, the images were resampled into isometric voxel of 1 × 1 × 1 mm^3^ by using B-spline interpolation, and the intensity histogram was discretized into a fixed bin width of 25. Second, radiomics features were extracted from the cropped cuboid VOIs which covered the tumor. Third, the radiomics features were scaled by using *z*-score normalization. 

The following radiomics features were then extracted by using the package “pyradiomics” (version 3.0.1, https://github.com/AIM-Harvard/pyradiomics, access on 17 November 2022): (1) 2D and 3D shape features (*n* = 14), (2) first-order statistics (*n* = 18), (3) gray level co-occurrence matrix-derived features (*n* = 24), (4) gray level run length matrix-derived features (*n* = 16), (5) gray level size zone-derived features (*n* = 16), (6) gray level dependence matrix-derived features (*n* = 14), (7) neighboring gray tone difference matrix features (*n* = 5), (8) features transformed by the wavelet on categories of (2)–(7) (*n* = 744). In total, 851 radiomics features were extracted. The terminology of the radiomics features is in line with the Image Biomarker Standardization Initiative [22] and the detailed definition of each feature can be found at https://pyradiomics.readthedocs.io/en/latest/features.html (access on 17 November 2022). 

### 2.5. Feature Selection, Model Development, and External Validation

To reduce the redundancy of the radiomics features, Spearman’s rank correlation analysis was adopted to evaluate their associations, with one feature randomly removed in the correlation pair with correlation coefficients >0.99. Random forest was then adopted to select the top 30 important features according to the Gini index (Figure 3). These 30 features were further fed into three classifiers, logistic regression (LR), support vector machine (SVM), and Adaboost to construct the radiomics models for predicting HCC histopathologic grades. When developing the prediction model, synthetic minority oversampling technique (SMOTE) was applied to overcome the unbalanced classification of the HCC histopathologic grades. To determine the hyper-parameter (the optimal number of features) of the model, a 10-fold cross-validation was performed in the development cohort. The developed models were then applied to Center 2 cohort to test their predictive performance. Above processes were achieved by open-source software with the build-in library scikit-learn 0.19 (https://scikit-learn.org) available at https://github.com/salan668/FAE (access date: 28 November 2022).

### 2.6. Statistical Analysis

The categorical variables were expressed as counts with percentages and compared by the chi-squared test or Fisher’s exact test as appropriate. Univariate regression analysis was applied to Center 1 cohort to detect the clinicopathological variable associated with the poorly differentiated HCC. The performance of the model was assessed by area under the receiver operating characteristic (AUC). The optimal cut-off value was determined by the Youden’s index. Sensitivity, specificity, accuracy, positive predictive value (PPV), and negative predictive value (NPV) at the optimal cut-off value were also calculated. The calibration ability of the model was visualized by the calibration plot, which intuitively compared the consistency between the model’s predicted probability and the actual probability. A Delong test was applied to evaluate the difference between the AUCs of different models. All statistical analyses were performed using R software (version 4.1.3, R Foundation for Statistical Computing, Vienna, Austria). A two-tailed *p*-value < 0.05 was regarded as statistically significant. 

## 3. Results

### 3.1. Basic Characteristics of Patient in the Two Cohorts

At Center 1, 265 eligible patients were included for model development, among which 86.0% were males and most patients were ≤55 years (70.2%). More than half (53.6%) of patients had a background of cirrhosis. There were 53.2% of patients with a tumor larger than 5 cm. AFP <400 ng/mL was seen in 58.5% of patients. The independent test cohort (Center 2) consisted of 138 patients, where 87.7% were males. Patients aged ≤55 years were observed in 56.5% of the test cohort. There were 47.1% of patients with a tumor larger than 5 cm. Around half (42.0%) of patients developed HCC on the basis of cirrhosis. A majority of patients (72.5%) had AFP < 400 ng/mL (Table 1). 

The incidence of poorly differentiated HCC was 15.1% and 15.9% at Center 1 and 2, respectively. Except for AFP at the Center 1 cohort, there were not significant differences in the other clinicopathological variables between poorly differentiated and non-poorly differentiated HCC groups in both cohorts. The basic characteristics of the patients at Center 1 and 2 are summarized in Table 1. 

### 3.2. Independent Clinical Predictor for Histopathological Grading

Univariable regression analysis of the association between the clinicopathological variables and the histopathologic grade of HCC in the development cohort only detected AFP to be significant, with an odds ratio of 2.75 (*p* < 0.05). Multivariable regression analysis was waived as only one significant clinicopathological variable was detected in univariate regression analysis (Table 2).

### 3.3. Feature Selection and Model Development 

Among the 851 extracted features, 502 features indicated good reproducibility (ICC ≥ 0.75). After the further removal of 137 redundant features (Spearman correlation coefficient > 0.99), 365 features were subjected to the random forest algorithm. After that, the top 30 important radiomics features evaluated by random forest were selected to develop three prediction models. The final features included in the models were determined by the learning curve, with 22, 22, and 10 in the LR, SVM, and Adaboost models, respectively (Appendix A). The key parameters in the model development are provided in Appendix A. 

### 3.4. Prediction Performance of the Radiomics Model 

The three radiomics models (LR, SVM, and Adaboost) yielded an AUC of 0.75 (95%CI: 0.68–0.83), 0.75 (95%CI: 0.68–0.83), and 0.93 (95%CI: 0.89–0.97), respectively, at the development cohort and of 0.70 (95%CI: 0.58–0.81), 0.67 (95%CI: 0.56–0.79), and 0.61 (95%CI: 0.47–0.74) at the test cohort. When incorporating AFP, the only significant clinicopathological variable, into these models, the AUCs of the combined models increased to 0.78 (95%CI:0.70–0.86), 0.78 (95%CI:0.70–0.85), and 0.94 (95%CI: 0.90–0.98), respectively, in the development cohort and to 0.71 (95%CI: 0.59–0.82), 0.69 (95%CI: 0.57–0.81), and 0.58 (95%CI: 0.45–0.72), respectively, in the test cohort, although the difference between the AUCs was not statistically significant (all *p* > 0.05). The LR+AFP model was therefore selected due to its best performance in the test cohort, which represents a promising model for HCC differentiation level prediction (the formula is provided in Appendix A). The ROCs of the combined LR+AFP model in both development and test cohorts are presented in Figure 4A. The calibration curve showed that the LR+AFP model had a good agreement between its predicted probability and the actual probability (Figure 4B,C). Detailed predictive performance in other dimensions, including accuracy, sensitivity, specificity, PPV, and NPV of these models at the development and test cohorts is summarized in Table 3. 

## 4. Discussion 

This study developed three radiomics models based on preoperative gadoxetic acid-enhanced MRI images for predicting the histopathologic grade of HCC in 265 patients. The predictive performance of the model using the logistic regression classifier outperformed the other two, having an AUC of 0.70 in an independent test cohort, and the AUC further increases to 0.71 when the model was combined with AFP. 

Radiomics has proven to be a powerful tool in the prediction of tumor differentiation, for instance, glioma, soft tissue sarcoma, and prostate cancer [23,24,25]. In the hepatobiliary field, several studies have explored the role of gadoxetic acid-enhanced MRI-based radiomics in the prediction of the HCC histopathologic grading [17,18,19]. A study constructed a radiomics model based on hepatobiliary phase images of gadoxetic acid-enhanced MRI using the logistic regression classifier yielded an AUC of 0.82 in the internal validation cohort [17]. In an early study, researchers conducted the textual analysis on the gadoxetic acid-enhanced MR images for predicting the HCC histopathologic grade, and the entropy showed the highest AUC (0.78) [18]. However, these studies were limited by the small sample size (<200) and the lack of an external validation cohort. Until now, only one study validated their radiomics model based on the hepatobiliary phase images in an independent test cohort [19]. That model showed the highest AUC of 0.70 in the test cohort—a result similar to ours. Yet, the external test cohort in that study consisted of only 28 cases [19]. 

The good performance of the gadoxetic acid-enhanced MRI-based model for histopathologic grading of HCC might be due to the underlying link between radiomics features and the biological behaviors of HCC. With the progression of HCC, the expression of OATPs decreases while the MRP2 remains consistent or increases, which results in a decreased absorption of gadoxetic acid [10]. Through quantifying the tumor uptake of gadoxetic acid, traditional studies have demonstrated a close association between HCC histopathologic grading and the alterations of the signal intensity or T1 relaxation time [26,27]. Radiomics features, the substantial imaging patterns extracted from the MR images, can better reflect the dedicated alterations of the image and would be also closely correlated with the tumor differentiation degrees. In addition, when tumors differentiate poorly and exhibit aggressive behaviors, the tumor’s internal environment may become more heterogeneous [18]. By capturing these heterogeneous textural patterns, radiomics features can well discriminate the histopathologic grades of HCC. Specifically, a majority of radiomics features included in the models in this study were wavelet related. The wavelet filter is a robust tool for obtaining a comprehensive spatial and frequency distribution to analyze specific imaging regions by combining low- and high-frequency signals [28]. 

Three classifiers were applied in this study, which resulted in the LR, SVM, and Adaboost-based radiomics models. LR is a traditional approach for modeling and is widely used in the biomedical field with good interpretability. SVM, as one of the most popular supervised machine learning techniques, is a powerful algorithm in the task of classification by detecting a line or a hyperplane to separate the samples into different classes [29]. By combining multiple weak learning classifiers via an iterative approach, Adaboost creates an ensemble stronger classifier and has an advantage in the accuracy improvement of the prediction [30]. Interestingly, the best performance was achieved by the LR model in our study, rather than by the two machine learning algorithms. A possible explanation might be that the performance of different classifiers was determined by the characteristics of the data. To put it in another way, only a classifier that fits the pattern of the data may show excellent predictive power. Under this context, a strategy of applying several classifiers for modeling may be useful to develop an accurate prediction model in the radiomics research field [31]. It is of note to point out that not always machine learning algorithms are superior to conventional classifiers in predicting clinical events. A recent systematic review compared the predictive performance between logistic regression and other machine learning algorithms based on 71 original studies [32]. The results showed that machine learning-based models do not outperform the logistic regression-based models [32]. 

As a well-established tumor biomarker, AFP has a high specificity in the early detection of HCC and is widely used for the diagnosis and surveillance of HCC [33]. In this study, AFP was the only significant clinicopathological variable associated with HCC differentiation levels. This finding was in line with previous reports [34,35]. Compared with AFP < 400 ng/mL, the odds ratio was 2.75 for AFP ≥ 400 ng/mL correlated with poorly differentiated HCC in this study. Recent evidence showed that a high level of AFP at baseline is also significantly associated with early tumor recurrence and poor prognosis of HCC in different clinical settings, such as after liver resection or liver transplantation [36,37]. These findings suggested the aggressive biological behavior of HCC with a higher AFP level [38,39].

For model development, a common strategy to improve predictive power is to incorporate risk factors from different dimensions [40]. In this study, when AFP was combined with the radiomics model, the AUC improved in both development and test cohorts, although the increment was not significant. In this multi-omics era, it is reasonable to integrate clinicopathological variables, radiomics, genomics, transcriptomics proteomics, and microbiomics variables to accurately predict the histopathologic grading of HCC and to achieve individualized treatment [41,42]. Ding et al. developed a model by combining clinicopathological variables, radiomics signature, and deep learning signature based on preoperative computed tomography images for predicting HCC differentiation levels [43]. The fused model had an AUC of 0.80 in an independent external cohort, which was significantly higher than models derived from each modality alone [43]. 

There are some limitations in this study. First, the data were retrospectively collected, and selection bias might be inevitable. In addition, the sample size of the test cohort was limited and only one test cohort was included. Additional external cohorts are required to further test the overfitting risk and generalizability of our model. Second, the difference in the predictive performance between radiological features or quantitative parameters based on signal intensity and the radiomics model was not compared in this study. There are strengths and weaknesses in these approaches in terms of subjectivity, complexity, and accuracy [27]. The comparison between them is interesting and clinically meaningful. Future studies can be designed to achieve this aim. Third, based on previous studies which demonstrated that the radiomics models based on the hepatobiliary phase outperformed other phases and sequences, only features derived from the hepatobiliary phase were employed in this study. Lastly, the number of the radiomics features included in the models was still partly subjectively determined as they were from the top 30 important features ranked by the random forest algorithm. The risk of overfitting might exist in our models (for example, the AUC difference of the Adaboost-based model between the development and test cohorts was big), although our feature selection strategy followed a “standard workflow” in a radiomics study. To date, there is not a well-established principle to define an ideal number of features for modeling, especially for different classifiers. Future studies are required to establish such a principle.

## 5. Conclusions

In conclusion, a radiomics model based on hepatobiliary phase images of preoperative gadoxetic acid-enhanced MRI was developed for predicting the histopathologic grade of HCC. Its good performance in the independent cohort indicates a promise in the preoperative prediction of HCC differentiation levels and in assistance of treatment management of HCC patients. 

## Figures and Tables

**Figure 1 diagnostics-13-00413-f001:**
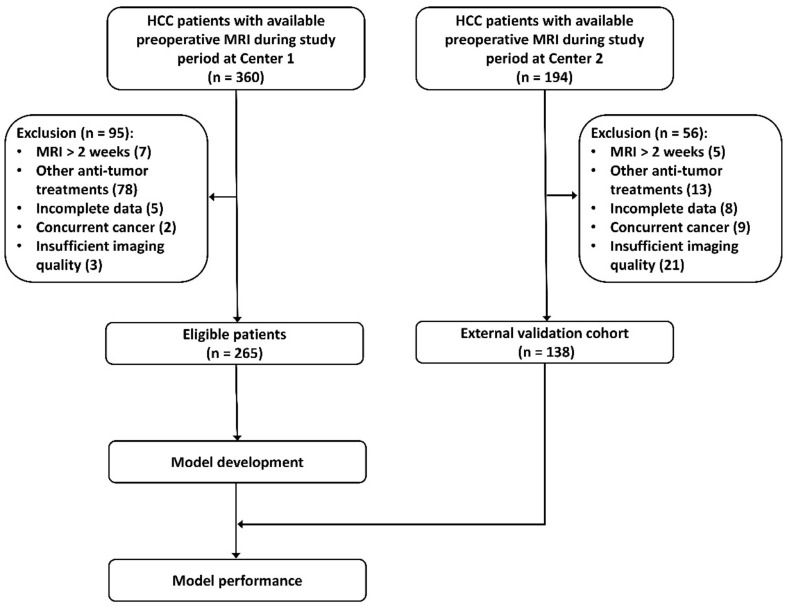
Process of patient selection at the two participating centers.

**Figure 2 diagnostics-13-00413-f002:**
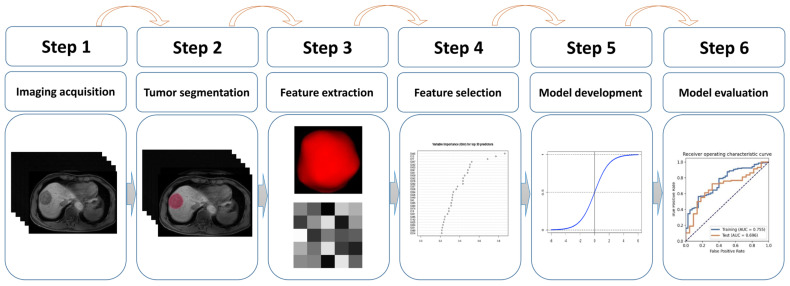
Workflow of the radiomics model development. A typical pipeline of the radiomics model development consists of six steps: imaging acquisition, tumor segmentation, feature extraction, feature selection, radiomics model development, and evaluation of the model.

**Figure 3 diagnostics-13-00413-f003:**
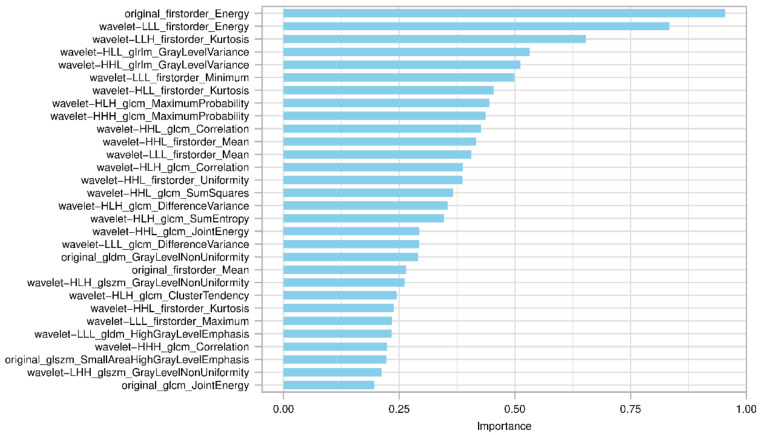
Top 30 radiomics features with importance evaluated by the Gini index in the random forest algorithm based on the development cohort. Gini Index is an effective measure of the impurity in the values of a dataset. In general, the higher the Gini index, the more important the feature is.

**Figure 4 diagnostics-13-00413-f004:**
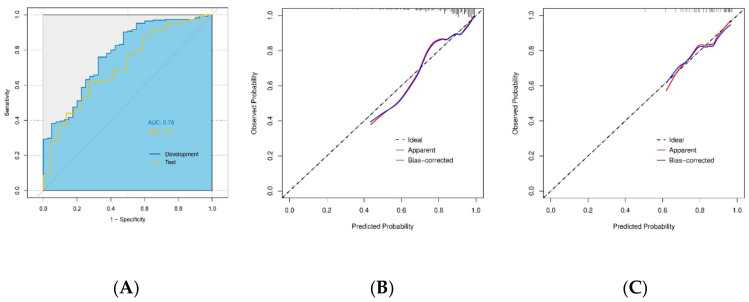
Predictive performance of the logistic regression radiomics model combined with alpha-fetoprotein (AFP) model at the model development cohort and the test cohort. (**A**) The combined model yielded an Area Under the Receiver Operating Characteristic curve of 0.78 in the development cohort and 0.71 in the test cohort. The calibration curve showed a good agreement between the combined model predicted probability and the actual probability in the development cohort (**B**) and the test cohort (**C**).

**Table 1 diagnostics-13-00413-t001:** Basic characteristics of the patients in the development and test cohorts.

	Center 1				Center 2			
	Overall (*n* = 265)	Poorly Differentiated HCC (*n* = 40)	Non-Poorly Differentiated HCC (*n* = 225)	*p* Value	Overall (*n* = 138)	Poorly Differentiated HCC (*n* = 22)	Non-Poorly Differentiated HCC (*n* = 116)	*p* Value
Gender:				0.343				1.000
Female	37 (14.0%)	8 (20.0%)	29 (12.9%)		17 (12.3%)	2 (9.09%)	15 (12.9%)	
Male	228 (86.0%)	32 (80.0%)	196 (87.1%)		121 (87.7%)	20 (90.9%)	101 (87.1%)	
Age (years):				0.334				0.617
≤55	186 (70.2%)	25 (62.5%)	161 (71.6%)		78 (56.5%)	14 (63.6%)	64 (55.2%)	
>55	79 (29.8%)	15 (37.5%)	64 (28.4%)		60 (43.5%)	8 (36.4%)	52 (44.8%)	
Etiology:				0.64				1.000
HBV	203 (76.6%)	29 (72.5%)	174 (77.3%)		112 (81.2%)	18 (81.8%)	94 (81.0%)	
Non HBV	62 (23.4%)	11 (27.5%)	51 (22.7%)		26 (18.8%)	4 (18.2%)	22 (19.0%)	
Cirrhosis:				1.000				0.725
Cirrhosis	142 (53.6%)	21 (52.5%)	121 (53.8%)		58 (42.0%)	8 (36.4%)	50 (43.1%)	
Non cirrhosis	123 (46.4%)	19 (47.5%)	104 (46.2%)		80 (58.0%)	14 (63.6%)	66 (56.9%)	
ALT (IU/L):				0.663				0.678
≤42	154 (58.1%)	25 (62.5%)	129 (57.3%)		105 (76.1%)	18 (81.8%)	87 (75.0%)	
>42	111 (41.9%)	15 (37.5%)	96 (42.7%)		33 (23.9%)	4 (18.2%)	29 (25.0%)	
AST (IU/L):				0.836				0.441
≤42	153 (57.7%)	22 (55.0%)	131 (58.2%)		106 (76.8%)	15 (68.2%)	91 (78.4%)	
>42	112 (42.3%)	18 (45.0%)	94 (41.8%)		32 (23.2%)	7 (31.8%)	25 (21.6%)	
Platelet (×109/L):				0.153				1.000
≤125	89 (33.6%)	9 (22.5%)	80 (35.6%)		29 (21.0%)	4 (18.2%)	25 (21.6%)	
>125	176 (66.4%)	31 (77.5%)	145 (64.4%)		109 (79.0%)	18 (81.8%)	91 (78.4%)	
ALBI grade:				0.213				0.738
Grade 1	120 (45.3%)	14 (35.0%)	106 (47.1%)		49 (35.5%)	9 (40.9%)	40 (34.5%)	
Grade 2	145 (54.7%)	26 (65.0%)	119 (52.9%)		89 (64.5%)	13 (59.1%)	76 (65.5%)	
MELD score:				0.629				0.589
≤9	256 (96.6%)	38 (95.0%)	218 (96.9%)		132 (95.7%)	22 (100%)	110 (94.8%)	
>9	9 (3.40%)	2 (5.00%)	7 (3.11%)		6 (4.35%)	0 (0.00%)	6 (5.17%)	
Tumor size (cm):				0.446				0.596
≤5	124 (46.8%)	16 (40.0%)	108 (48.0%)		73 (52.9%)	10 (45.5%)	63 (54.3%)	
>5	141 (53.2%)	24 (60.0%)	117 (52.0%)		65 (47.1%)	12 (54.5%)	53 (45.7%)	
AFP (ng/mL):				0.006 *				0.204
<400	155 (58.5%)	15 (37.5%)	140 (62.2%)		100 (72.5%)	13 (59.1%)	87 (75.0%)	
≥400	110 (41.5%)	25 (62.5%)	85 (37.8%)		38 (27.5%)	9 (40.9%)	29 (25.0%)	

Note: * *p* < 0.05. AFP—alpha fetoprotein; ALBI grade—albumin-bilirubin grade; ALT—alanine aminotransferase; AST—aspartate transaminase; HBV—hepatitis B virus; HCC—hepatocellular carcinoma; MELD score—model for end-stage liver disease score.

**Table 2 diagnostics-13-00413-t002:** Univariable regression analysis of the clinicopathological variables associating with histopathologic grading of HCC in the development cohort.

Clinicopathological Variable	OR	95%CI	*p* Value
Gender (Female vs. male)	0.59	0.25–1.41	0.24
Age (≤55 vs. >55 years)	1.51	0.75–3.05	0.25
Etiology (Non HBV vs. HBV)	0.77	0.36–1.65	0.51
Cirrhosis (Non cirrhosis vs. cirrhosis)	0.95	0.48–1.86	0.88
ALT (≤42 vs. >42 IU/L)	0.81	0.40–1.61	0.54
AST (≤42 vs. >42 IU/L)	1.14	0.58–2.24	0.70
Platelet (≤125 vs. >125 × 10^9^/L)	1.90	0.86–4.19	0.11
ALBI grade (Grade 1 vs. 2)	1.65	0.82–3.33	0.16
MELD score (≤9 vs. >9)	1.64	0.33–8.19	0.55
Tumor size (≤5 vs. >5 cm)	1.38	0.70–2.75	0.35
AFP (<400 vs. ≥400 ng/mL)	2.75	1.37–5.50	<0.001 *

Note: * *p* < 0.05. AFP—alpha fetoprotein; ALBI grade—albumin-bilirubin grade; ALT—alanine aminotransferase; AST—aspartate transaminase; CI—confidence interval; HBV—hepatitis B virus; HCC—hepatocellular carcinoma; MELD score—model for end-stage liver disease score; OR—odds ratio.

**Table 3 diagnostics-13-00413-t003:** Summary of the model performance in the development and test cohorts.

	Model	AUC (95%CI)	Cut-Off Value	Accuracy	Sensitivity	Specificity	PPV	NPV
Development cohort	LR	0.75 (0.68–0.83)	0.56	0.61	0.56	0.85	0.95	0.26
	SVM	0.75 (0.68–0.83)	0.41	0.81	0.85	0.58	0.92	0.40
	Adaboost	0.93 (0.89–0.97)	0.50	0.85	0.85	0.88	0.97	0.51
	LR+AFP	0.78 (0.70–0.86)	0.83	0.75	0.76	0.68	0.93	0.33
	SVM+AFP	0.78 (0.70–0.85)	0.84	0.73	0.73	0.73	0.94	0.33
	Adaboost+AFP	0.94 (0.90–0.98)	0.73	0.91	0.92	0.85	0.97	0.67
Test cohort	LR	0.70 (0.58–0.81)	-	0.72	0.72	0.68	0.92	0.32
	SVM	0.67 (0.56–0.79)	-	0.68	0.68	0.68	0.92	0.29
	Adaboost	0.61 (0.47–0.74)	-	0.75	0.80	0.45	0.89	0.30
	LR+AFP	0.71 (0.59–0.82)	-	0.64	0.62	0.72	0.92	0.27
	SVM+AFP	0.69 (0.57–0.81)	-	0.80	0.86	0.45	0.89	0.38
	Adaboost+AFP	0.58 (0.45–0.72)	-	0.77	0.83	0.45	0.89	0.33

Note: AFP—alpha fetoprotein; AUC—area under the receiver operating characteristic curve; CI—confidence interval; LR—logistic regression; NPV—negative predictive value; PPV—positive predictive value; SVM—support vector machine.

## Data Availability

The original contributions presented in the study are included in the article/Appendix A. Further inquiries can be directed to the corresponding authors.

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
