# Peer review of "Development and External Validation of a Radiomics Model Derived from Preoperative Gadoxetic Acid-Enhanced MRI for Predicting Histopathologic Grade of Hepatocellular Carcinoma"

_diagnostics, 2023, doi:10.3390/diagnostics13030413_

Round 1

Reviewer 1 Report

This study developed and validated a gadoxetic acid-enhanced MRI-based radiomics model for predicting histological grade in patients with single HCC who underwent surgical resection treatment. The manuscript was well-written and very interesting. 

The evaluation for overfitting is important in Radiomics studies. When the number of selected features is too many, it can lead to overfitting. Please re-check whether there is overfitting and correct for potential overfitting.

Author Response

Dear Editor and reviewer:

We would like to express our appreciation for the work and insightful comments by the reviewer, helping us to improve our manuscript further. Response to the comments is listed point-by-point as follows (in red). The modified parts according to these comments are tracked in the main manuscript.

Reviewer#1:

This study developed and validated a gadoxetic acid-enhanced MRI-based radiomics model for predicting histological grade in patients with single HCC who underwent surgical resection treatment. The manuscript was well-written and very interesting. 

Response: Thank you very much for your positive feedback!

The evaluation for overfitting is important in Radiomics studies. When the number of selected features is too many, it can lead to overfitting. Please re-check whether there is overfitting and correct for potential overfitting.

Response: This is really a crucial issue when developing a radiomics model. To date, there is not a well-established principle to define an ideal number of features in a (radiomics) prediction model. We assume that the number must be a trade-off between model simplicity and the model discriminative ability. The number of the radiomics features in the three classifiers was 22, 22, and 10 respectively in our study, which is a moderate number of features in a radiomics model. We double-checked our study workflow and the feature selection process to detect any potential overfitting. Although our feature selection strategy followed a “standard workflow” in a radiomics study (ICC analysis, correlation analysis, random forest for ranking feature importance, and 10-fold cross-validation), the risk of overfitting was relatively small for our LR+AFP model. However, potential overfitting may exist in the Adaboost and Adaboost + AFP models as the AUC gaps between the development and test cohorts were big. Besides, we only tested our model in one independent external cohort and the overfitting risk for the LR+AFP model cannot be excluded. We added a relevant statement in the manuscript to remind the readers to interpret our model with caution (P14 L58-65). 

The following discussion about feature number determination is for your information (due to the length consideration it is not included in the manuscript):

Traditionally, the “10 EPV” (events per variable) principle was usually adopted in the logistic regression analysis. However, this criterion has been questioned [1]. Our previous systematic review, which summarized 22 studies about the radiomics models for predicting microvascular invasion in HCC, showed that the ratio of events to the number of the selected radiomics features (E/F ratio) ranged between 0.7 and 35.5 (median: 4.2) (the median number of selected features was 15, with a range 2 to 74) [2]. Our ongoing systematic review on this topic (radiomics models for predicting pathological grade of HCC) included 12 original studies, and we found that the E/F ratio ranged from 1.8 to 29 (median: 3.7) with the number of the selected radiomics features varied between 1 and 50 (median: 11).

The number of radiomics features included in our model was 22, 22, and 10 in the three classifiers and the E/F ratio was 1.8, 1.8, and 4 respectively, which was consistent with previous studies. Still, there might be a risk of overfitting in our model. As there is not any principle for an ideal number of features included in a model, future studies can be designed to evaluate the effect of the number of the radiomics feature and the model performance and to provide some recommendations for model developers.

Thank you very much for your insightful comments!

References:

  1. Courvoisier DS, Combescure C, Agoritsas T, Gayet-Ageron A, Perneger TV. Performance of logistic regression modeling: beyond the number of events per variable, the role of data structure. J Clin Epidemiol. 2011 Sep;64(9):993-1000. doi: 10.1016/j.jclinepi.2010.11.012.
  2. Wang Q, Li C, Zhang J, Hu X, Fan Y, Ma K, Sparrelid E, Brismar TB. Radiomics Models for Predicting Microvascular Invasion in Hepatocellular Carcinoma: A Systematic Review and Radiomics Quality Score Assessment. Cancers (Basel). 2021 Nov 22;13(22):5864. doi: 10.3390/cancers13225864.

Reviewer 2 Report

General comments: Well written manuscript. Timely topic of predicting histopathologic grade of HCC with gadoxetic acid enhanced MRI.

Large sample size with one model cohort and testing cohort.

Specific comments:

Page 2 lines 73/74: missing units for 10-40--should be minutes

Page 11 line 47: Change APF to AFP

Page 13 lines 91-93: "Radiomics features, which are imaging patterns high throughput extracted from the dedicatedly altered..."  This sentence is confusing and should be reworded.

Author Response

Dear Editor and reviewer:

We would like to express our appreciation for the work and insightful comments by the reviewer, helping us to improve our manuscript further. Response to the comments is listed point-by-point as follows (in red). The modified parts according to these comments are tracked in the main manuscript.

Reviewer#2

General comments: Well written manuscript. Timely topic of predicting histopathologic grade of HCC with gadoxetic acid enhanced MRI.

Large sample size with one model cohort and testing cohort.

Response: Thank you very much for this comment!

Specific comments:

Page 2 lines 73/74: missing units for 10-40--should be minutes

Response: The unit “minutes” has been added. Sorry for this superficial mistake.

Page 11 line 47: Change APF to AFP

Response: The typo “APF” has been replaced by “AFP”.

Page 13 lines 91-93: "Radiomics features, which are imaging patterns high throughput extracted from the dedicatedly altered..."  This sentence is confusing and should be reworded.

Response: It has been rephrased as “Radiomics features, the substantial imaging patterns extracted from the MR images, can better reflect the dedicated alterations of the image and would be also closely correlated with the tumor differentiation degrees.”

Thank you very much for the insightful comments and constructive suggestions!